# Huntingtin Interacting Proteins and Pathological Implications

**DOI:** 10.3390/ijms241713060

**Published:** 2023-08-22

**Authors:** Li Liu, Huichun Tong, Yize Sun, Xingxing Chen, Tianqi Yang, Gongke Zhou, Xiao-Jiang Li, Shihua Li

**Affiliations:** Guangdong Key Laboratory of Non-Human Primate Research, Key Laboratory of Central Nervous System Regeneration (Ministry of Education), Guangdong-Hongkong-Macau Institute of CNS Regeneration, Jinan University, Guangzhou 510623, China; liuli66@stu2021.jnu.edu.cn (L.L.); tonghuichun@stu2021.jnu.edu.cn (H.T.); syz98@stu2020.jnu.edu.cn (Y.S.); chenxingxing80@aliyun.com (X.C.); yangtianqi12138@stu2021.jnu.edu.cn (T.Y.); zgk00@stu2022.jnu.edu.cn (G.Z.); xjli33@jnu.edu.cn (X.-J.L.)

**Keywords:** huntingtin, protein interaction, Huntington’s disease, polyglutamine

## Abstract

Huntington’s disease (HD) is caused by an expansion of a CAG repeat in the gene that encodes the huntingtin protein (HTT). The exact function of HTT is still not fully understood, and previous studies have mainly focused on identifying proteins that interact with HTT to gain insights into its function. Numerous HTT-interacting proteins have been discovered, shedding light on the functions and structure of HTT. Most of these proteins interact with the N-terminal region of HTT. Among the various HTT-interacting proteins, huntingtin-associated protein 1 (HAP1) and HTT-interacting protein 1 (HIP1) have been extensively studied. Recent research has uncovered differences in the distribution of HAP1 in monkey and human brains compared with mice. This finding suggests that there may be species-specific variations in the regulation and function of HTT-interacting proteins. Understanding these differences could provide crucial insights into the development of HD. In this review, we will focus on the recent advancements in the study of HTT-interacting proteins, with particular attention to the differential distributions of HTT and HAP1 in larger animal models.

## 1. Introduction

Huntington’s disease (HD) is an autosomal dominant inherited neurodegenerative disease. The primary cause of HD is the CAG expansion mutation of the huntingtin (HTT) gene, which is located on chromosome 4p16 in humans, resulting in the clinical manifestation of HD [1]. In normal healthy individuals, the number of CAG repeats is typically less than 35, whereas individuals with HD have 36 or more CAG repeats. The range of CAG repeats in the majority of HD patients is usually between 40 and 50 and causes adult-onset symptoms [2]. Although HD patients with a higher number of repeats tend to develop symptoms at an earlier age, this correlation is more obvious in juvenile patients carrying the larger repeats (>65 CAGs) when compared with the majority of HD patients who harbor the intermediate repeats (40–50 CAGs). The HTT in juvenile HD patients has more than 60 CAG repeats, which typically leads to the onset of the disease during childhood or adolescence [1,2].

The CAG repeats in the HTT gene encode a polyglutamine (polyQ) tract, such that expanded CAG repeats result in the expansion of a polyQ repeat in HTT. The pathological feature of HD is the preferential loss of medium spiny neurons (MSN) in the striatum, and neurodegeneration spreads to other regions of the brain as the disease progresses. Clinically, HD is characterized by emotional and mental disturbances, involuntary movements, and cognitive decline. As the disease advances, individuals gradually lose their ability to speak, perform daily routine tasks, and think. HD continues to progress over approximately 10 to 20 years, eventually leading to the death of the patient.

Protein-protein interactions are crucial for numerous cellular functions, and any malfunction or disruption in these interactions has been implicated in various pathological conditions [3]. HTT engages in interactions with a wide range of proteins. It serves to maintain an intracellular protein network and potentially plays vital roles in intracellular transportation or acts as a cytoskeleton scaffold [4,5]. Mutations in HTT can not only affect its own function but also impair the function of other proteins that interact with it. Therefore, studying the proteins that interact with HTT is of significant importance [4]. Investigating HTT-interacting proteins can enhance our understanding of the pathogenic mechanism of HD and may open up new avenues for the treatment of the disease.

In this review, we will provide an overview of the proteins that interact with HTT, which can be influenced by the polyQ numbers or whose function may be important for HD development and have been extensively investigated. Additionally, we will concentrate on the recent discoveries related to huntingtin-associated protein 1 (HAP1), with the objective of gaining new insights into the mechanisms underlying the pathogenesis of HD.

## 2. Structure of HTT and Expression of Mutant HTT

HTT, a soluble protein comprised of 3144 amino acids (348 kDa), is expressed throughout the body, with higher levels found in the central nervous system and testis [6]. The N-terminal 17 amino acids, also referred to as the N17 region, have been identified as a critical region involved in the localization, aggregation, and toxicity of HTT [7]. Following the N17 region is the polyQ region, which contains up to 35 CAG repeats in individuals without Huntington’s disease (HD). Subsequently, there is a polyproline-rich region [8] that interacts with various proteins containing the SH3 domain [9]. After the polyproline-rich region, there are clusters of HEAT repeats (α-helix-loop-α-helix motif) that play a crucial role in mediating interactions between HTT and other proteins (Figure 1).

Nuclear localization and nuclear export signals have been identified within HTT protein, respectively [10,11,12,13]. The amino acid sequence of HTT contains significant post-translational modification sites, such as ubiquitination/sumoylation, phosphorylation, palmitoylation, acetylation, and proteolysis [14]. Many of these modification sites are located within or near four regions known as PEST domains, which consist of proline-rich, glutamic acid, serine, threonine, and predicted cleavage sites [15].

The expansion of CAG repeats in exon 1 of the HTT gene leads to the translation of a mutant HTT protein with an enlarged polyglutamine (PolyQ) tract in its N-terminal region. The major toxic form of HTT is believed to be the N-terminal fragments carrying the expanded polyQ [16], which can result from aberrant splicing [17,18,19,20] or proteolytic processes [21,22,23,24,25,26]. These mutant N-terminal fragments can enter the nucleus and become retained there through self-aggregation or oligomerization. Large aggregates form inclusion bodies, and HTT aggregation can disrupt the balance of interacting proteins and may be toxic. The toxic effects of mutant HTT include transcriptional dysregulation, synaptic dysfunction, mitochondrial toxicity, and impaired axonal transport [3,4].

By understanding the proteins that interact with HTT and contribute to its toxic effects, researchers can gain valuable insights into the molecular pathways involved in HD. The identification of HTT-interacting proteins involved in HD pathogenesis is crucial for defining the underlying mechanisms of the disease and finding potential targets for treatment.

The aberiviations are: HPIP: huntingtin protein interacting proteins; Herp: homocysteine-induced endoplasmic reticulum protein; SP1: specificity protein 1; GAPDH: glyceraldehyde-3-phosphate dehydrogenase; HAP1: huntingtin-associated protein 1; Rhes: ras homolog enriched in the striatum; Hip1: huntingtin interacting protein 1; KMO: kynurenine 3-Monooxygenase; HYPK: huntingtin interacting protein K; VCP/p97: ATPase valosin-containing protein; SIP: siah1-interacting protein; RAC1: rac family small GTPase 1; Hap40: huntingtin-associated protein 40.

## 3. HTT Interacting Proteins

The functions of HTT are still not fully understood. The interaction between HTT and other proteins influences the state, function, and other aspects of HTT and its interacting proteins, leading to related physiological changes [27]. Mutant HTT exists in at least two forms: diffuse (soluble) and aggregated (insoluble), both of which are cytotoxic [28]. The soluble form of mutant HTT can be cleared by the cellular metabolic system, such as the proteasome. However, the aggregated insoluble form of mutant HTT has reduced solubility, is highly compact, and is difficult to clear by the proteasome. As a result, it accumulates and aggregates over time [29].

HTT-interacting proteins can bind to either normal or mutant HTT. The binding affinity of certain HTT-interacting proteins increases as the polyQ number (number of glutamine repeats) in mutant HTT increases, while other proteins show reduced binding affinity with increasing polyQ number Table 1 and Figure 1. We will select a few examples for further discussion.

### 3.1. Proteins Binding to HTT Enhanced by Poly Q Expansion

#### 3.1.1. Huntingtin-Associated Protein 1 (HAP1)

HAP1, a neuronal protein, was initially discovered as the first Huntingtin-interacting protein through the use of the yeast two-hybrid system (Y2H) [30]. Since its discovery, HAP1 has been extensively studied as one of the key proteins involved in HTT interactions. In rodents, HAP1 exists in two alternative spliced subtypes, which differ only at the very C-terminal end (amino acids 579–599 in HAP1A and 579–629 in HAP1B) [30]. Notably, HAP1 exhibits a stronger binding affinity to HTT with an expanded glutamine repeat compared with wild-type HTT [30,31]. Although several studies have replicated our findings (30), the mechanism by which polyQ expansion can alter protein-protein interactions remains to be defined. It is possible that the polyQ expansion alters the protein conformation, thereby increasing the binding of more proteins to the HTT-N-terminal portion. However, the increased binding of HAP1 to mutant HTT appears to provide protection to neuronal cells, as demonstrated by significant cell death in mice and monkey tissues when HAP1 is knocked down in the presence of mHTT [32,33].

Rodent HAP1 is predominantly expressed in the hypothalamus and has been found to interact with various proteins involved in intracellular trafficking, receptor function, transcription factors, cytoskeleton proteins, and even form dimers and octamers through self-interaction [30,34]. Its role in intracellular transport/trafficking and receptor endocytosis has been well established [22,31]. Recent studies have also highlighted the protective role of HAP1 against mutant HTT (mHtt) accumulation, as knockdown of HAP1 in the presence of mHtt leads to neuronal cell death [32,33]. Both HAP1 and HTT are implicated in intracellular trafficking in neuronal cells, as they both interact with the microtubule machinery [27,34]. However, HAP1 is also involved in other cellular processes such as gene transcriptional regulation, signal transduction, and autophagy [35,36,37]. It is plausible that the interaction between HTT and HAP1 plays a crucial role in various cellular functions, and disruption of the HTT mutation affects the function of both proteins. It is worth mentioning that HAP1 was found to be localized on a previously identified cytoplasmic structure called the “stigmoid body” [30,38,39]. Our subsequent studies have discovered that Ahi1 and Dcaf7/WDR68 are also localized to this unique structure with HAP1 [40,41]. Initially, this membrane-less condensed structure was observed using hPAX-P2 [38], an antibody derived from human placenta antigen, and later confirmed to react with the HAP1 C-terminal amino acid sequences using the hPAX-P2S antibody [42]. These findings suggest that hPAX-P2S is identical to the STB-constituted HAP1 and that the HAP1-induced/inclusion bodies correlate with the hPAX-P2-immunoreactive STBs previously identified in the brain [39,42]. While HAP1A alone, but not HAP1B, was found to form the stigmoid-like structure in transfected cells [39], the exact function of HAP1 in the stigmoid body remains to be investigated.

#### 3.1.2. Glyceraldehyde-3-Phosphate Dehydrogenase (GAPDH)

GAPDH, also known as glyceraldehyde-3-phosphate dehydrogenase, exhibits both glyceraldehyde-3-phosphate dehydrogenase and nitrosylase activities [43], making it involved in glycolysis and other cellular functions [44]. The reports on the interaction between mutant HTT (mHTT) and GAPDH are somehow inconsistent. Some studies have indicated that the polyQ repeats in the HTT protein specifically interact with GAPDH, and this interaction is enhanced with longer polyQ stretches [45,46]. On the other hand, other research has shown that mutant HTT binds to GAPDH through its interaction with Siah1, an E3 ligase, resulting in cell death [25]. Furthermore, mHtt with different N-terminal polymers disrupts GAPDH-mediated micro-mitophagy, leading to the accumulation of damaged mitochondria in cells. This impairment of micro-mitophagy may contribute to the pathological events observed in HD [47].

#### 3.1.3. Transcription Factor Sp1 (Sp1)

Sp1 is a transcription factor that can activate or repress transcription in response to physiological and pathological stimuli. It binds to GC-rich DNA motifs with high affinity and regulates the expression of numerous genes involved in various processes, including cell growth, apoptosis, cellular differentiation, and immune responses [48]. Previous studies have shown that HTT interacts with Sp1, and the expansion of polyglutamine enhances this interaction [49,50]. The binding of mHTT to Sp1 impairs its function, resulting in the inhibition of nuclear Sp1 binding to the promoter of the nerve growth factor receptor. This leads to reduced transcriptional activity in cultured cells or disrupts the coordinated transcriptional activity of Sp1 and TAFII130 in HD mouse models or patients with HD [50].

Another study investigating astroglial-specific mHtt expression demonstrated that mHtt reduces the expression of the glutamate transporter in astrocytes. This is achieved by increasing its binding to Sp1, thereby reducing the binding of Sp1 to the glutamate transporter promoter. Consequently, this causes glutamate excitotoxicity and neuronal death [51]. Other studies have found functional Sp1-responsive elements in the HTT gene promoter region. The expression of Sp1 enhances the transcription of HTT, while inhibiting Sp1-mediated transcription reduces the expression of mHtt [52]. Additionally, increased SP1 activity has been reported in cellular and transgenic HD mouse models [53]. The downstream effects of the interaction between Sp1 and mHtt are complex. The imbalance in Sp1-mediated HTT transcription, combined with the adverse effects of mHtt on downstream SP1-dependent gene expression, may contribute to the pathogenesis of HD at different disease stages.

#### 3.1.4. Homocysteine-Induced Endoplasmic Reticulum Protein (Herp)

Herp, an E3 ubiquitin ligase, is an integral membrane protein regulated by the ER stress response pathway. It is involved in the ubiquitination and degradation of many mutant proteins [54]. It was found that Herp can bind to the overexpressed HTT N-terminus in HTT-transfected N2a cells, and this interaction is enhanced with the polyQ-expanded HTT N-terminal fragments [55]. Mutant HTT transgenic mice show increased Herp expression in their brain regions, and Herp reduces the cytotoxicity of mHtt by inhibiting its aggregation and promoting its degradation [55].

Herp’s N-terminal acts as a molecular chaperone, inhibiting protein aggregation, while its C-terminal acts as an E3 ubiquitin ligase, promoting the degradation of misfolded proteins through UPP [54]. Understanding the mechanisms by which Herp regulates HTT degradation and aggregation could have significant implications for the development of therapeutic strategies for protein misfolding diseases, such as HD. By targeting Herp or its associated pathways, it may be possible to enhance the clearance of misfolded proteins and alleviate the cytotoxic effects associated with their accumulation. Further research in this area is warranted to explore the full potential of Herp as a therapeutic target in HD.

### 3.2. Proteins Binding to HTT Inhibited by PolyQ Expansion

#### 3.2.1. Huntingtin-Interacting Protein 1 (HIP1)

Apart from HAP1, HIP1 is also one of the most extensively studied proteins that interact with HTT. The binding affinity between HTT and HIP1 decreases as the number of polyglutamine residues exceeds the pathogenic threshold [56,57]. HIP1 is a membrane-associated protein that plays a role in clathrin-mediated endocytosis and intracellular protein trafficking [58]. HIP1 exists in multiple variants, each with its own specific function, and they also work together to fulfill their roles.

In HD, the disruption of the normal HTT-HIP1 interaction may result in cytoskeletal defects and impaired membrane receptor trafficking in the brain, ultimately leading to neuronal cell death [58]. In addition to its interaction with HTT, HIP1 also interacts with other proteins and has been implicated in various medical conditions, including tumor genesis such as lung cancer and leukemia [59]. HIP1 is also involved in the development of rheumatoid arthritis by modulating the PDGFR and Rac1 signaling pathways [60]. Furthermore, it has been reported that patients with epilepsy exhibit a deletion of the chromosome segment that contains HIP1 [61]. It is worth noting that HTT-interacting proteins have multiple other functions that may be affected by mHTT.

#### 3.2.2. Kynurenine 3-Monooxygenase (KMO)

Kynurenine 3-Monooxygenase (KMO) is another protein that interacts with normal HTT, but its interaction with mHTT is reduced [62]. Studies have demonstrated that KMO physically interacts with soluble HTT exon 1 protein fragments within living cells, primarily at the mitochondrial outer membrane [62]. Specifically, the expansion of the pathogenic polyglutamine tract in HTT leads to the formation of intracellular protein inclusions, which disrupts the interaction with KMO [62]. KMO is located in the mitochondrial outer membrane and is responsible for catalyzing the synthesis of 3-hydroxykynurenine from L-kynurenine [63]. It plays a crucial role in the degradation of tryptophan through the kynurenine pathway (KP), controlling the balance between the formation of kynurenic acid (KYNA) and the synthesis of downstream neurotoxic metabolites [64]. The neuroactive metabolites produced by the tryptophan-degrading enzyme kynurenine pathway (KP) are implicated in the pathophysiological processes of neurodegenerative diseases, including HD [64,65]. Furthermore, deletion of KMO in the R6/2 HD mice has been reported to rescue inflammatory reactions in these mice [66]. Given its significance in cellular processes, KMO’s interaction with HTT may regulate mitochondrial function. Further studies are required to dive into the intricacies of KMO-HTT interaction biology and comprehend their implications for HD pathogenesis.

## 4. Interacting Protein that Can Enhance the mHTT Toxicity

HTT-interacting proteins originate from various sources, some of which can reduce mHTT toxicity and protect neuronal cells. For instance, HAP1 (32, 33) has been shown to have these beneficial effects. On the other hand, there are proteins such as Rhes that enhance mHTT toxicity and lead to cell death [67]. Rhes belongs to the Ras superfamily of small GTPases and functions as an E3 ligase, facilitating the attachment of small ubiquitin-like modifier (SUMO) to its targets. It has been found that Rhes initiates mHTT sumolation, resulting in the production of more soluble mHtt, which is believed to be more toxic than the aggregated form of mHtt [68].

Rhes is primarily found in the striatum, and knocking down Rhes has been shown to protect striatal neuronal cell death in a 3′-NP-induced HD mouse model [69]. Additionally, Rhes has been found to regulate autophagy activity in an HD knock-in mouse model [70]. Given that striatal neuronal death is selective in HD and Rhes is enriched in this brain region with a function that is crucial for protein aggregation-related diseases, Rhes emerges as a strong candidate for linking to HD pathogenesis. However, it is important to note that all the intriguing findings regarding the interaction between Rhes and mHtt have been generated using HD mouse models or cell models. It would be interesting to investigate whether these results can be replicated in large animal models of HD.

## 5. HTT Interacting Proteins that Can Alter mHtt Aggregates

Many proteins that interact with HTT have been reported to be capable of altering mHtt aggregation. In this discussion, we will briefly explore some of these HTT-interacting proteins. One such protein is HTT interacting protein K (HYPK), which belongs to the AAA ATPase family. HYPK binds to the first 17 amino acids of the N-terminus of HTT (HTT-N17), inhibiting the formation of mHtt aggregates and reducing neuronal cell death caused by mHtt [71,72]. HYPK’s regulation of mHtt aggregation may be attributed to its chaperoning activity [72]. Another protein, VCP, acts as a decomposing enzyme that binds to mHtt and reduces its aggregation [73]. VCP’s ability to break down protein aggregates stems from its depolymerizing enzyme function, which may guide the mHtt protein for degradation [73].

Additionally, Siah1-interacting protein (SIP) also plays a role in regulating mHtt aggregation. SIP is a multi-domain and multi-ligand protein that interacts with various proteins and plays important roles in the cell [74]. Siah1 is a central component of a multiprotein E3 ubiquitin ligase complex, and SIP binds to Siah1 within this complex [86]. Wild-type SIP increases mHtt ubiquitination, reduces mHtt protein levels, and decreases mHtt aggregation [75]. However, an increase in SIP dimerization in HD MSN leads to a decrease in SIP’s function in degrading mHtt through the ubiquitin-proteasome pathway, resulting in an increase in mHtt aggregation [75]. Protein interactions have an impact on the aggregation of mHtt, which, in turn, may affect the pathology of HD. Therefore, further research should be conducted to investigate the mechanisms of these protein interactions with mHtt, which could potentially lead to new directions in treating protein aggregation-related neurological disorders.

## 6. Proteins with Impaired Function When Interacting with mHtt

Proteins with known functions are valuable targets for studying the mechanisms of HD when their functions are altered by interacting with mHTT. When a protein interacts with mHtt, its own function can also be affected. For example, Xu et al. identified Vimentin as the preferential interaction factor of mHtt [76]. Vimentin normally forms a cage-like structure around aggregates in neural stem cells (NSCs) to maintain cellular proteostasis [77]. However, in HD NSCs, mHtt impairs the ability of Vimentin to form a cage around the aggregates in response to proteotoxic stress, resulting in a decline in the proliferation and neurogenesis of the NSCs [78].

Another example of mHtt affecting the function of an interacting protein is illustrated by its ability to retard the reactivity of Rac1 to BDNF-stimulated growth activity in human NSCs and a mouse HD model [79]. Rac1, a member of the Rho family of GTPases, is an intracellular transducer known to regulate multiple signaling pathways that control cytoskeleton organization, transcription, and cell proliferation [80]. Rac1 is a downstream target of growth factor receptor/PI3 kinase activity and is critical for actin-dependent membrane remodeling in response to growth signaling [81]. HTT normally regulates Rac1 activity as part of a coordinated response to growth factor signaling, but this function is altered by mHtt in the early stages of HD [79].

Beclin 1, another example of impaired function due to interaction with mHtt, is a key component of autophagy [82,87]. Beclin 1 is sequestered by mHtt, and this interaction affects the degradation of long-lived protein aggregates through autophagy. HTT has been reported to potentially serve as a scaffold for selective autophagy [7], but mHtt may lose this function and aberrantly interact with different components of the autophagy system, resulting in impaired autophagy function in various HD models [4,82,87,88,89,90].

## 7. New Insights in HAP1

HAP1, the first HTT-associated protein identified through yeast two-hybrid screening, has been extensively studied by numerous researchers across various model systems. Notably, several differences have been observed between rodents and large animal models [91,92,93]. As a result, the distribution of HAP1 in rodents and primates has been investigated, along with its potential function in primate brains [33]. In this paragraph, we aim to provide a comprehensive overview of the latest findings for the distribution of HAP1 and HTT. These findings could contribute to a better understanding of HAP1’s role in HD and its potential implications for therapeutic interventions.

(1)HAP1 in the primate brains

Human HAP1 (hHAP1) was discovered a few years later through similarity PCR cloning, following the discovery of rat HAP1. It shares 62% identity with rat HAP1 across its entire sequence and 82% amino acid identity in the putative HTT-binding region [94]. Human HAP1 also binds to HTT, and this binding is enhanced by increased polyQ numbers. Interestingly, a previous study found that the expression level of hHAP1 is low in the human brain striatum and is significantly decreased in the HD brain striatum [94]. Some researchers have questioned its role in HD pathogenesis, as rodent Hap1 is primarily enriched in the hypothalamus, which is not the most affected brain region in HD [30,95].

However, since Hap1 has shown a protective effect in HD KI mouse striatum [32], it is worth further investigating whether HAP1 is distributed differently in primate brains compared with rodent brains and if it indeed participates in HD pathogenesis. Mutation of HTT has been found to be fully responsible for causing HD [96], and HTT mRNA and protein were reported to be ubiquitously distributed throughout the body in rodents and humans [96,97,98,99,100]. Therefore, selective neurodegeneration cannot be fully explained solely by the mutation in HTT. The selective neurodegeneration in HD raises the possibility of mHtt affecting neuronal protein function through abnormal protein-protein interactions and causing specific neuronal cell death [101,102,103]. In HD, the medium spiny neurons (MSN) in the striatum are the most vulnerable neuronal cells, and brains from patients with HD show severe loss of MSN in the striatum at the late stages of HD [101,102].

In the phenomenon of selective neurodegeneration, when a mutant protein is expressed ubiquitously in all cells, there are two possible explanations for the pathogenesis of the disease. The first is that the specific mutant protein is enriched in the brain region that is most affected and the enriched expression of mutant protein causes the selective cell death. The second is that a protein or proteins with an important function in this brain region are being affected by the mutant protein. In the case of HD, the mutant protein is found to be distributed in all cells, with the highest expression observed in the brain and testis [96,97,98,99,100].

Following the discovery that the mutant huntingtin (HTT) protein causes HD and is ubiquitously distributed in the body, many researchers began searching for HTT-interacting proteins. One such protein is HAP1, which is enriched in the brain and was the first reported HTT-interacting protein [30]. Subsequently, many other HTT-interacting proteins have been identified, some of which are neuronally-enriched while others are not. Additionally, some interactions are enhanced by polyQ expansion, while others do not show this enhancement [3,4]. Many of these studies have been conducted using rodent models, human cell lines, and drosophila models for confirmation. Only a few interactions have been discovered through human tissue studies [88,89,90].

Given that large animal models have physiological functions, anatomical structures, metabolic pathways, and behavioral phenotypes more similar to humans than rodents [91,92,93], it is important to consider the investigation of disease pathogenesis, particularly in relation to brain diseases, using large animal models.

Our recent study compared the expression patterns of primate HAP1 and HTT in monkey brains. The results showed that HAP1 and HTT are both expressed in similar brain areas, including the striatum, cortex, hippocampus, and hypothalamus [33] and (Figure 2b). In contrast to the highly enriched expression of Hap1 in the rodent hypothalamus [30,33] and (Figure 2a), the levels of HAP1 and HTT expression in different regions of the monkey brain are relatively similar, with higher abundance observed in the striatum, cortex, and hippocampus [33] and as shown in (Figure 2b). Immunofluorescent staining of the monkey brain striatum and cortex revealed that both Hap1 and HTT are expressed in the cytoplasm of the same cells [33]. Furthermore, immunoprecipitation of HAP1 with rabbit anti-hHAP1 from the monkey brain cortex resulted in the significant co-precipitation of HTT. In contrast, there was negligible Hap1 in the mouse cortex that could be precipitated by the HAP1 antibody. These findings indicate that there is a higher presence of HAP1 and HTT in the monkey cortex, where they interact with each other in vivo [33].

The expression patterns of HAP1 in monkey brains differ significantly from those in mice [33]. Additionally, the parallel expression of HAP1 and HTT in primate brains emphasizes the importance of studying these proteins in primate brains. The expression pattern of HAP1 in the monkey brain suggests its involvement in the pathogenesis of HD, and the interaction between mHtt and HAP1 may impact the function of these proteins, thereby affecting the pathology of HD. This is especially evident when both proteins are expressed in the same cells in the most vulnerable brain regions [33,104,105]. It would be interesting to conduct a comprehensive study on HAP1 and HTT using a relevant HD model, such as the HD knock-in pig model [106], or a more precise monkey HD model that can replicate the selective neuropathology observed in human patients.

(2)HAP1 is neuroprotective in monkey tissue

Mutation of a protein can result in either the loss of its own function or the gain of a toxic function through binding to other proteins, which could be its regular interactor or a protein that normally does not interact with the mutant protein. It has been found that both HAP1 and HTT participate in intracellular transport [5,31,34]. The consequence of this interaction could be that the mutant protein binds more tightly to the interactor and causes its loss of function. Indeed, research has reported impaired intracellular transport in the presence of mHtt [107,108].

Proteins that interact with HTT may have a protective role in the intracellular environment, and their deletion may be detrimental to the cells. For instance, in the presence of mutant HTT, deletion of Hap1 caused striatal cell death in the HD KI 140Q mouse model [32], even though the expression level of Hap1 in the mouse brain is not the highest in the striatum [30,39]. This result suggests that Hap1 is protective for mouse striatal neuronal cells, which agrees with previous RNA expression study results [109].

In a large animal model that more closely resembles humans in terms of physiology, anatomy, and metabolism, certain proteins might distribute differently and have functions different from those in rodents. Our study showed that HAP1 distributes differently in the monkey brain compared with rodent brains. When we knocked out HAP1 using CRISPR/Cas9 in the monkey brain slice culture with the expression of mHtt, it also caused a significant amount of cell death [33]. This result further confirms that HAP1 indeed participates in HD pathogenesis in the monkey brain.

(3)Other functions of HAP1 in humans

Up to this point, seven isoforms of human HAP1 (hHAP1) transcripts have been documented in the NCBI gene database. These isoforms share a conserved N-terminal region but diverge from the middle of the transcript, suggesting potential alternative splicing events. As a result, these isoforms give rise to seven distinct HAP1 proteins. It has been observed that hHAP1 transcripts are predominantly expressed in the brain and stomach, with higher levels detected in the stomach compared with the brain. Researchers have demonstrated that hHAP1 is localized in the human digestive tract [110].

HAP1 has been implicated in various types of cancer, including gastric cancer [111], acute lymphoblastic leukemia (ALL) [35], breast cancer [112,113], and lung cancer [114]. These findings suggest a potential role for hHAP1 in regulating cell growth and cell death pathways. However, due to the existence of seven isoforms of hHAP1, the distribution of these isoforms within the human body and their specific functions in different systems remain unclear. It is important to note that although rodent HAP1 has been extensively studied, it only presents with two isoforms [30]. Therefore, the distribution and function of rodent HAP1 cannot be directly extrapolated to explain the expression and function of hHAP1.

## 8. Conclusions and Future Remarks

HD is caused by a poly-CAG expansion mutation in the HTT gene. HTT was discovered 30 years ago through positional cloning [96]. However, the exact function of HTT is still not fully understood. Knocking out HTT in mice resulted in embryonic death [83,115,116], while humans with large CAG repeat mutations can still be born alive [84]. To understand the selective neuronal cell death in HD, many researchers have conducted studies to identify HTT-interacting proteins that may explain the vulnerability of striatal neurons. Over the past 30 years, the majority of HTT-interacting proteins were found to bind to the N-terminus of the HTT protein, while only a few were found to bind to the very C-terminal, such as HAP40. HAP40 was initially identified to bind and associate with the carboxyl-terminal of HTT through protein co-purification [85]. A recent cryo-EM study revealed that HAP40 binds to the HEAT repeats in HTT, thereby stabilizing the HTT-HAP40 complex [117]. Other studies have demonstrated that HAP40 interacts with mHtt exon1 and that its binding to HTT is dependent on the polyQ repeat expansion [118]. However, the association between HAP40 and HD pathogenesis remains unclear, with conflicting reports on the matter. One study suggests a huntingtin-dependent decrease in HAP40 protein levels in both HD cell lines and mouse models [119]. However, another report indicates that in HD drosophila and mouse models, HAP40 only stabilizes HTT and does not affect HTT exon1 toxicity [120]. To better understand the role of HAP40 in HD pathogenesis, further rigorous in vivo studies are required to determine the true function of the HTT-HAP40 complex association. The findings from mouse studies and the identification of interacting proteins have greatly contributed to our understanding of the biological function of HTT. Some of these proteins are specific to neurons, such as HAP1 [30], while others are not. Only a small number of proteins were found to be highly expressed in the striatum, such as Rhes [67].

Most of the HTT-interacting proteins were found in mouse or cell models of HD. So far, none of the HTT-interacting proteins has been shown clearly to participate in the HD pathogenesis or is the only modulator that is directly involved in the HD manifestation clinically in patients. Only a few studies using data from human HD samples have shown that HTT-interacting proteins can modify the age of onset of HD [121,122]. This could be due to the fact that mHtt affects the functions of many interacting proteins, and HD is a result of the abnormal function of all the affected proteins, which aligns with the theory of a toxic function mutation. Alternatively, HD could result from mHtt losing its own function, as suggested by a few studies that have shown mHtt affecting animal development at an early stage [123,124,125]. However, most adult patients with HD do not manifest clinical symptoms until middle age, suggesting that the loss of normal function of HTT may not be the major player in the disease development of the majority of patients with adult-onset HD. The loss of HTT function scenario might be seen in Juvenile HD, these patients have an early onset of the clinical symptoms, and the clinical progress is much faster than the adult-onset of HD. A third possibility could be that the interacting proteins behave differently in different animal species due to the significant differences between humans and rodents [126,127]. For example, both HAP1 and HTT are important for the proliferation and differentiation of mouse neuro-stem cells (mNSC), but they function differently in human NSC (hNSC) [33]. The loss of HAP1 and HTT will impact the cell proliferation and neural development of mNSC, and it will also affect the expression of many gene families related to neuronal development in mice [33]. However, in hNSC, the loss of HAP1 does not affect neurogenesis and does not lead to significant changes in gene transcription [33], indicating that HAP1 in primate neurons may not be important for the early development of neurons but may be crucial for maintaining the function of mature neurons. On the other hand, HTT deficiency can affect hNSC neuronal differentiation and development and have a widespread impact on gene transcription [33].

Although HAP1 and HTT function differently in primate neurons for neuronal development, both HAP1 and HTT are expressed in the monkey brain in a very similar pattern and associate with each other. Knocking down HAP1 in monkey brain slice culture resulted in significant neuronal loss in the presence of mHtt. This result suggests that dysfunction of HAP1 may promote mHtt-mediated neurotoxicity in the primate brain, which supports the view that HAP1 dysfunction is associated with age-dependent neurodegeneration in the presence of mHTT in HD [33].

The distinct brain distribution of rodent HAP1 and primate HAP1, as well as the findings from the comparative study on HTT and HAP1 in mouse and primate brains, highlight an important point. Despite the high degree of similarity between rodent and primate genomes, individual proteins can exhibit different distribution patterns and functions in their respective brains. This discrepancy may underlie the significant differences observed between large animal and small animal models when studying human brain disorders [126,127]. Therefore, it is crucial to consider the disparities between primate and rodent animal models when selecting an appropriate model for studying brain diseases or analyzing the results of human disease studies in animal models. The primate brain HAP1 study serves as just one example of the differences between humans and rodents. Conducting a more detailed comparison of HAP1 or other promising HTT-interacting proteins’ distribution between human and primate brains would further validate the current findings.

## Figures and Tables

**Figure 1 ijms-24-13060-f001:**
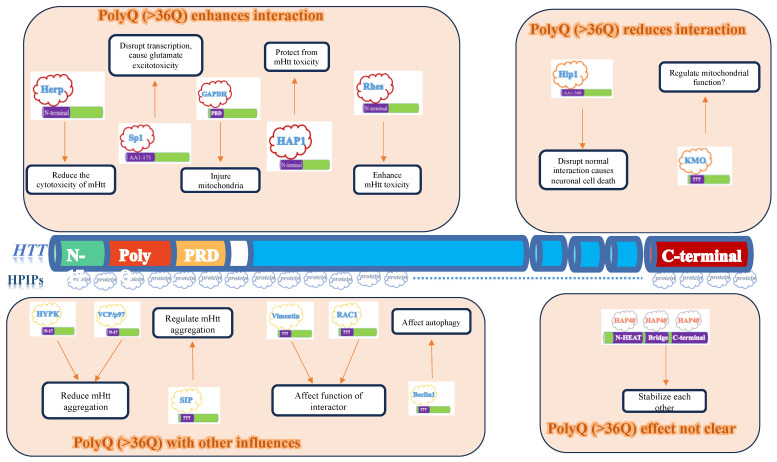
Illustration of HTT interacting proteins discussed in this manuscript.

**Figure 2 ijms-24-13060-f002:**
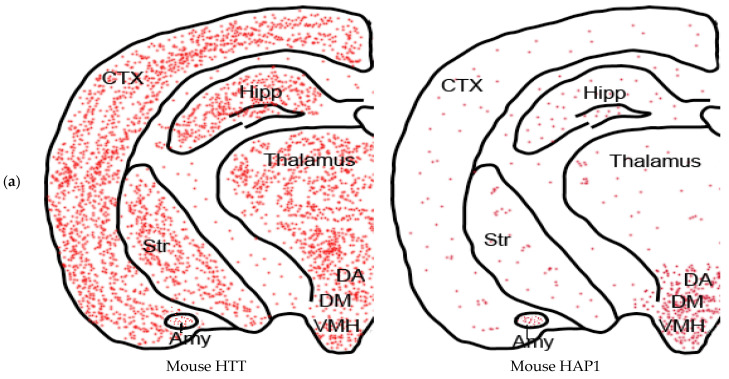
Schematic illustration of the expression pattern of HTT and HAP1 in mouse and primate brains. Red dots represent the densities of HTT or HAP1 in the mouse (**a**) and primates brain (**b**) areas. Aberiviations are: CTX: cortex, Str: Striatum; Amy: Amygdala; Hipp: Hippacampus; DA: dorsal hypothalamus; DM: dorsomedial hypothalamus; VHM: ventral medial hypothalamus; Pu: putamen; Cd: caudate nuclears; Gp: globus pallidus; Hypo: Hypothalamus; LV: lateral ventricle.

**Table 1 ijms-24-13060-t001:** List of the HTT interacting proteins with its possible function, which suggest that may play HD pathogenesis.

Name	Function	Region in Htt for Binding	PolyQ-Length Influence	Identification Method	References
Huntingtin associated protein 1 (HAP1)	multiple functions	N-terminal	Enhances	Y2-H	[30,31,32,33,34,35,36,37,38,39,40]
GAPDH	Glycolitic enzyme	Polyproline	Enhances	Affinity chromatography	[41,42,43,44,45]
Sp1	Transcription activator	Amino acid 1–171	Enhances	CO-IP, Y2-H	[46,47,48,49,50,51]
Homocysteine-induced endoplasmic reticulum protein (Herp)	Ubiquitin ligase	N-terminal	Enhances	Co-IP	[52,53]
Huntingtin-interacting protein 1 (Hip1)	Clathrin-mediated endocytosis and trafficking	Amino acid 1–540	Decreases	Y2-H	[54,55,56,57,58,59]
Kynurenine 3-Monooxygenase (KMO)	Flavin monooxygenase	Unknown	Decreases	BiFC, Co-IP	[60,61,62,63,64]
Rhes	Small gtpase/E3 ligase	N-terminal	Increases	Co-IP	[65,66,67,68]
HYPK	Chaperone protein	N-terminal 17 amino acid domain	Unknown	Co-IP	[69,70]
ATPase valosin-containing protein (VCP/p97)	Facilitate the process of protein polyubiquitination	N-terminal 17 amino acid domain	Unknown	IP	[71]
Siah1-interacting protein(SIP)	Ubiquitin ligase	Unknown	Unknown	Co-IP, Autophagy test	[72,73,74]
Vimentin	Stabilize intracellular structure	Unknown	Unknown	Co-IP, Proximity ligation assay (PLA)	[75,76]
RAC1 (Rac family small GTPase 1)	Plasma membrane-associated small GTPase	Unknown	Unknown	IP	[77,78,79]
Beclin 1	A key initiator of autophagy	Unknown	Unknown	IP	[80,81,82]
HAP40	Stabilize HTT protein	C-terminal	N/A	Co-purification	[83,84,85]

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
