# Peer review of "Huntingtin Interacting Proteins and Pathological Implications"

_ijms, 2023, doi:10.3390/ijms241713060_

Round 1
Reviewer 1 Report
The manuscript is well-written and could be acceptable for publication in IJMS. However, this reviewer has some concerns.
1. Line 55-56, HTT is primarily expressed in the central nervous system and testis, particularly in the cerebral cortex of the brain. However, a recent paper showed the detailed distribution of HTT using a rabbit monoclonal antibody against HTT (Hartlage-Rübsamen et al., acta neuropathol commun 7, 79, 2019). They reported that, employing a monoclonal anti-HTT antibody directed against the HTT mid-region and using brain tissue of three different mouse strains, prominent immunoreactivity in a number of brain areas, particularly in cholinergic cranial nerve nuclei. Please clarify this issue.
2. Line 106, HAP1 exhibits a stronger binding affinity to HTT with expanded glutamine repeat than wildtype HTT. If the authors clarify why HAP1 has a stronger binding affinity to HTT with expanded glutamine repeat, would it be helpful for the reader?
3. It has been reported that HAP1 is an essential component of the stigmoid body (STB). HAP1 has two isoforms, HAP1A, and B, especially, HAP1A gives rise to produce STB. The functions of HAP1 and STB could be similar. The authors should include the relationships of HAP1 and STB as new insights into HAP1 functions.
4. A schematic diagram demonstrating the HTT, its interactor, and its possible functions would be better for understanding the review.
English language is fine.
Author Response
Reviewer 1
- Line 55-56, HTT is primarily expressed in the central nervous system and testis, particularly in the cerebral cortex of the brain. However, a recent paper showed the detailed distribution of HTT using a rabbit monoclonal antibody against HTT (Hartlage-Rübsamen et al., 2019). They reported that, employing a monoclonal anti-HTT antibody directed against the HTT mid-region and using brain tissue of three different mouse strains, prominent immunoreactivity in a number of brain areas, particularly in cholinergic cranial nerve nuclei. Please clarify this issue.
We would like to express our gratitude to the reviewer for addressing the concern regarding the localization of wild-type HTT. We have thoroughly examined the mentioned reference (Hartlage-Rübsamen et al., 2019) during our research on the topic. However, we had reservations about the findings presented in the paper. It is important to note that the study relied on a single HTT antibody from Abcam, which, although validated for specificity using human cell lines lacking HTT, exhibited discrepancies in the western blot analysis. The western blot analysis showed a smaller “HTT” band than expected and some non-specific bands. Furthermore, the reference paper did not provide any western blot data demonstrating the antibody's ability to detect rodent HTT in brain homogenates, despite showing staining in the brains. Considering that humans and rodents are distinct species with different physiological environments, an antibody that recognizes a human protein in a cell line does not necessarily ensure recognition of the same protein in a different species. Consequently, we do not deem this reference appropriate for studying HTT distribution before other researcher confirm its use in another similar study. Conversely, previous studies on HTT distribution have employed well-characterized antibodies, resulting in consistent outcomes across multiple experiments (Sharp et al., 1995; Landwehrmeyer et al., 1995). Additionally, the distribution of HTT transcripts has been extensively investigated using in situ hybridization and Northern blotting (Li et al., 1993; Strong et al., 1993), These works have established that HTT is widely expressed and enriched in the brain and testis.
- Line 106,HAP1 exhibits a stronger binding affinity to HTT with expanded glutamine repeat than wildtype HTT. If the authors clarify why HAP1 has a stronger binding affinity to HTT with expanded glutamine repeat, would it be helpful for the reader?
This is a very intriguing question. However, despite several studies successfully replicating our previous findings (Li et al., 1995), the mechanism for how polyQ expansion can alter protein-protein interactions remains to be defined. The prevalent theory is that expanded polyQ repeats can alter protein conformation, especially for truncated N-terminal fragments. The evidence to support this theory includes the facts that truncated N-terminal fragments are able to misfold and form aggregates, and many HTT-interacting proteins are found to bind N-terminal region of mutant HTT (Wanker et al.,2019). However, the increased binding of HAP1 to mutant HTT protected the neuronal cells since knocking down of HAP1 in the presence of mHTT resulted significant cell death in mice and monkey tissues. We have added this paragraph in the section:
Although several studies have replicated our findings (30), the mechanism by which polyQ expansion can alter protein-protein interactions remains to be defined. It is possible that the polyQ expansion alters the protein conformation, thereby increasing the binding of more proteins to the HTT-N-terminal portion. However, the increased binding of HAP1 to mutant HTT appears to provide protection to neuronal cells, as demonstrated by significant cell death in mice and monkey tissues when HAP1 is knocked down in the presence of mHTT.
- It has been reported that HAP1 is an essential component of the stigmoid body (STB). HAP1 has two isoforms, HAP1A, and B, especially, HAP1A gives rise to produce STB. The functions of HAP1 and STB could be similar. The authors should include the relationships of HAP1 and STB as new insights into HAP1 functions.
We appreciate this good suggestion. Currently, there is a lack of comprehensive research regarding the function and composition of the stigmoid body. Therefore, our understanding of this structure is limited. Our early study showed that HAP1A, but not HAP1B, is able to form stigmoid-like structures in transfected cells (Li et al., 1998), and other studies have demonstrated the localization of the hPAX-P2, Hap1, Ahi1, and Dcaf7/WDR68 within the stigmoid body by immuno-electro-microscopic and immunohistochemistry studies (Li et al., 1995, Sheng et al., 2008; Xiang et al., 2017). We have included the following discussion in the revision:
HAP1 was found to be localized on a previously identified cytoplasmic structure called the "stigmoid body" (Li et al., 1998, Shinoda et al., 1993). Our subsequent studies have discovered that Ahi1 and Dcaf7/WDR68 are also localized to this unique structure with HAP1 (Sheng et al., 2008; Xiang et al., 2017). Initially, this membrane-less condensed structure was observed using hPAX-P2, an antibody derived from human placenta antigen, and later confirmed to react with the HAP1 C-terminal amino acid sequences using the hPAX-P2S antibody. These findings suggest that hPAX-P2S is identical to the STB-constituted HAP1 and that the HAP1-induced/inclusion bodies correlate with the hPAX-P2-immunoreactive STBs previously identified in the brain (Fujinaga et al., 2007). While HAP1A alone, but not HAP1B, was found to form the stigmoid-like structured in transfected cells (Li et al., 1998), the exact function of HAP1 in the stigmoid body remains to be investigated."
- A schematic diagram demonstrating the HTT, its interactor, and its possible functions would be better for understanding the review.
We have provided a schematic diagram to illustrate the different HTT interacting proteins on page 4.

Reviewer 2 Report
A rather outdated review focusing on mostly older manuscript and not really “recent” advances.
The Hap40 section is all wrong, Hap40 does not bind the carboxyl-terminus of huntingtin only, cryoEM studies show HAP40 bind both halves of huntingtin. It does not only bind to wild type HTT. Recent manuscripts show HAP40 is co-translated and the levels of HAP40 and HTT and inter-dependent.
“So far, none of the HTT-interacting proteins has been shown clearly to participate 409 in the HD pathogenesis or is the only modulator that is directly involved in the HD man- 410 ifestation. “
Again, outdated. This review entirely entirely ignores the role of huntingtin in DNA repair, and interactors, which has morphed into a major research focus in this field.
Figure 1: Why are they trying to re-publish figures already in press??
Minor points
“ is located on the fourth chromosome in humans, resulting in the clinical manifestation of HD” line 25
This is more accurate to say chromosome 4, or 4p16.
“It is important 30 to note that the earlier the disease onset, the longer the CAG repeat tends to be.”
This is not accurate at all. There is a tendency to follow this rule, but CAG length alone is a poor predictor of age onset. This is the point of HD GWAS studies.
Lines 32-25, they need to mention cognitive symptoms and depression.
“HTT, a soluble protein consisting of 3144 amino acids (348 kDa), is primarily ex- 55 pressed in the central nervous system and testis, particularly in the cerebral cortex of the 56 brain [6]. “
No, not true, HTT is expressed in every cell in the human body and is essential for any cell type survival.
“Cytoplasmic retention signals and nuclear export signals have been identified at the 64 N-terminus and C-terminus of HTT, respectively.” These needs primary references.
“The major toxic form of HTT is believed to be the N-terminal fragments carrying the 72 expanded polyQ, which can result from aberrant splicing [12] or proteolytic processes 73 [13].”
This is outdated. The proteolysis hypothesis has not been studied much in the last 10 years and the aberrant splicing hypothesis is unique to one manuscript, from one lab, and is not seen in massive datasets of mRNA deep sequencing of the HTT transcript.
Kynurenine 3-Monooxygenase (KMO): it is very clear from studies 5 years ago onward this is likely not relevant to HTT or HD.
Terms like “polyQ” and “N-terminal are slang. They should use the correct term of polyglutamine and amino-terminal. C-terminal is slang, the correct term is carboxyl-terminal.
Author Response
Reviewer 2
A rather outdated review focusing on mostly older manuscript and not really “recent” advances.
The manuscript does highlight the latest advancements in the field, particularly the differential expression of HAP1 in primate and mouse brains, which suggest that the HAP1 might function differently in different spices. However, the manuscript also includes a substantial amount of previous
publications for discussion of the potential involvement of HTT interacting proteins in HD pathogenesis. Thus, we have changed the title to “Huntingtin Interacting Proteins and Pathological Implications”.
The Hap40 section is all wrong, Hap40 does not bind the carboxyl-terminus of huntingtin only, cryoEM studies show HAP40 bind both halves of huntingtin. It does not only bind to wild type HTT. Recent manuscripts show HAP40 is co-translated and the levels of HAP40 and HTT and inter-dependent.
HAP40 was first found to be associated with the HTT carboxyl-terminal by protein co-purification, and a structure study showed that HAP40 and full length HTT are in the same complex and stabilize HTT in a cryo-EM study (Guo et al., 2018). We have cited these findings in the manuscript. However, whether
HAP40 is protective or can exacerbate HD is controversial, because some reports show that HAP40 level is decreased in HD (Harding et al., 2021) and others show that HAP40 is significantly increased in HD (Xu et al., 2022). As for the interaction regions in HTT and HAP40, HAP40 was also found to interact with the mHTT exon1 (Harding et al., 2021). We have discussed HTT-HAP40 interaction as follows in the revised manuscript (line 411-421):
HAP40 was initially identified to bind the carboxyl-terminal of HTT through protein co-purification [114]. A recent cryo-EM study revealed that HAP40 binds to the HEAT repeats in HTT, thereby stabilizing the HTT-HAP40 complex [115]. Other studies have demonstrated that HAP40 interacts with mHtt exon1 and that its binding to HTT is dependent on the polyQ repeats expansion [116]. However, the association between HAP40 and HD pathogenesis remains unclear. One study suggests a huntingtin-dependent decrease in HAP40 protein levels in both HD cell lines and mouse models [117]. Another report indicates that in HD drosophila and mouse models, HAP40 only stabilizes HTT and does not affect HTT exon1 toxicity [118]. To better understand the role of HAP40 in HD pathogenesis, further rigorous in vivo studies are required to determine the true function of the HTT-HAP40 complex association.
“So far, none of the HTT-interacting proteins has been shown clearly to participate in the HD pathogenesis or is the only modulator that is directly involved in the HD manifestation.
It is true that there is no literature showing that manipulating a single HTT interacting protein can treat HD or significantly alter the behaviors of HD animal models. This is in line with the fact that mutant HTT interacts with multiple proteins to cause synergistic toxicity. Thus, discussion of each HTT
interacting proteins for their potential involvement in HD pathogenesis in our manuscript would be helpful for understanding the roles of HTT interacting proteins in HD.
Again, outdated. This review entirely ignores the role of huntingtin in DNA repair, and interactors, which has morphed into a major research focus in this field.
HTT is a large protein with complex functions that still remain to be fully investigated. Our manuscript focuses on HTT interacting proteins and aim to understand how mutant HTT alters the associations with these HTT interacting proteins, an important pathological process in HD. The interactions of HTT with DNA and other subcellular organelles are also likely involved in HD, which would be better described in an independent review.
Figure 1: Why are they trying to re-publish figures already in press??
We have replaced this figure with a new figure to indicate species-dependent differences in HAP1 expression. Please see figure 2 on page 10
Minor points
“ is located on the fourth chromosome in humans, resulting in the clinical manifestation of HD” line 25
This is more accurate to say chromosome 4, or 4p16.
Thanks for point out this, we have changed the words to Chromosome 4p16.
“It is important to note that the earlier the disease onset, the longer the CAG repeat tends to be.”
This is not accurate at all. There is a tendency to follow this rule, but CAG length alone is a poor predictor of age onset. This is the point of HD GWAS studies.
We are not sure which GWAS study the reviewer is referring to. However, in the adult HD cases that carry intermediate repeats between 40-55, the negative correlation of the repeat length with early disease onset is not obvious as the larger repeats (>65) found in juvenile patients. We have rephrased the statement (line 27-30) as follows:
Although,HD patients with a higher number of repeats tend to develop symptoms at an earlier age, this correlation is more obvious in juvenile patients carrying the larger repeats (>65 CAGs) when compared with the majority of HD patients who harbor the intermediate repeats (40-55 CAGs)."
Lines 32-25, they need to mention cognitive symptoms and depression.
We have added this sentence in the paragraph (line36-37) as follows:
Clinically, HD is characterized by emotional and mental disturbances, involuntary movements, and cognitive decline.
“HTT, a soluble protein consisting of 3144 amino acids (348 kDa), is primarily expressed in the central nervous system and testis, particularly in the cerebral cortex of the 56 brain [6]. “
No, not true, HTT is expressed in every cell in the human body and is essential for any cell type survival.
We have modified this sentence (Lin 55-56) as follows:
HTT, a soluble protein comprised of 3144 amino acids (348 kDa), is expressed throughout the body, with higher levels found in the central nervous system and testis”.
“Cytoplasmic retention signals and nuclear export signals have been identified at the N-terminus and C-terminus of HTT, respectively.” These needs primary references.
We have modified the sentence as “Nuclear localization and nuclear export signals have been identified within HTT protein, respectively” and added the following references for this sentence (line 64-65, and references 10-13).
Xia, J.; Lee, D.H.; Taylor, J.; Vandelft, M.; Truant, R. Huntingtin contains a highly conserved nuclear export signal. Hum Mol Genet. 2003 12(12):1393-403. doi: 10.1093/hmg/ddg156.
Desmond, C.R.; Atwal, R.S.; Xia, J.; Truant, R. Identification of a karyopherin β1/β2 proline-tyrosine nuclear localization signal in huntingtin protein. J Biol Chem. 2012, 287(47):39626-33. doi: 10.1074/jbc.M112.412379.
Zheng, Z.; Li, A.; Holmes, B.B.; Marasa, J.C.; Diamond, M.I. An N-terminal nuclear export signal regulates trafficking and aggregation of Huntingtin (Htt) protein exon 1. J Biol Chem. 2013, 288(9):6063-71. doi: 10.1074/jbc.M112.413575.
Maiuri, T.; Woloshansky, T.; Xia, J.; Truant, R. The huntingtin N17 domain is a multifunctional CRM1 and Ran-dependent nuclear and cilial export signal. Hum Mol Genet. 2013, 22(7):1383-94. doi: 10.1093/hmg/dds554.
“The major toxic form of HTT is believed to be the N-terminal fragments carrying the expanded polyQ, which can result from aberrant splicing or proteolytic processes.
This is outdated. The proteolysis hypothesis has not been studied much in the last 10 years and the aberrant splicing hypothesis is unique to one manuscript, from one lab, and is not seen in massive datasets of mRNA deep sequencing of the HTT transcript.
Because N-terminal HTT fragment formation is important for HD pathogenesis, we believe that the important references should be cited here to indicate how they can be produced, despite no new literature for potentially new mechanisms.
Kynurenine 3-Monooxygenase (KMO): it is very clear from studies 5 years ago onward this is likely not relevant to HTT or HD.
Recent studies have highlighted the significance of KMO in relation to HD pathogenesis (Bondulich et al., 2021, Swaih, et al., 2022; Fathi et al., 2022). However, the exact role of KMO in HD pathogenesis has not been fully elucidated. We have emphasized the need for additional rigorous in vivo studies to delve into the complexities of KMO-HTT interaction biology and better understand their implications for HD pathogenesis in the previous submitted manuscript. (Please see line 213)
Comments on the Quality of English Language
Terms like “polyQ” and “N-terminal are slang. They should use the correct term of polyglutamine and amino-terminal. C-terminal is slang, the correct term is carboxyl-terminal.
Thanks for pointing out these. When we first describe these terms in the paper, we indicate their full-names and abbreviations, which have been widely used in many publications in the reputable journals.
